# Available Strategies for the Management of Andean Lupin Anthracnose

**DOI:** 10.3390/plants11050654

**Published:** 2022-02-28

**Authors:** César E. Falconí, Viviana Yánez-Mendizábal

**Affiliations:** 1Departamento de Ciencias de la Vida, Carrera de Ingeniería Agropecuaria, Universidad de las Fuerzas Armadas (ESPE), Av. General Ruminahui s/n, Sangolqui 171103, Ecuador; 2Facultad de Ingeniería y Ciencias Aplicadas, Carrera de Ingeniería en Agroindustrias, Universidad de las Américas, Quito 170503, Ecuador

**Keywords:** Andean lupin, anthracnose, integrated management, plant protection, dry heat, UV-C, UV-B, *Bacillus subtilis*, biological control, ontogenic resistance, chemical control

## Abstract

The lupin (*Lupinus mutabilis* Sweet) is a legume domesticated and cultivated for more than 4000 years by the pre-Hispanic cultures of the Andean zone. Due to its good taste and protein content, the lupin seed contributes significantly to the food and nutritional security of the Andean population. However, lupin is susceptible to diseases, and of these, anthracnose is the most devastating as it affects the whole crop, including leaves, stems, pods, and seeds. This review focuses on available strategies for management of lupin anthracnose from sowing to harvest. Seed disinfection is the primary anthracnose management strategy. Seed treatment with fungicides reduces transmission from seed to seedling, but it does not eradicate anthracnose. Attention is given to alternative strategies to limit this seed-borne pathogen as well as to enhance plant resistance and to promote plant growth. For anthracnose management in the field, integrated practices are discussed that encompass control of volunteer plants, lupin ontogenetic resistance, and rotation of biocontrol with chemical fungicides at susceptible phenological stages. This review covers some local experiences on various aspects of anthracnose management that could prove useful to other the groups focusing on the problem.

## 1. Introduction

*Lupinus mutabilis* Sweet is a Fabaceae used as food since pre-Inca times in the Andean countries. It is characterized by its high nutritional value, atmospheric nitrogen fixation in soil, and adaptation to various climatic conditions with minimal soil and water requirements [1,2]. In the Andean areas of Peru, Ecuador, and Bolivia, lupin constitutes one of the main sources of food and income for indigenous populations [2]. Since the 1960s, researchers have attempted to solve some of the main agronomic problems related to planting, fertilization, crop rotation, and crop management to improve productivity [3], postharvest quality, and marketing [4]. However, the food industry demands continuous production of high quality and uniform lupin seeds to sustain their processes [5]. Diseases, and particularly anthracnose, have a direct negative effect on production and quality of lupin seed [5,6]. Therefore, it is important at this time to review the available anthracnose management strategies to synthesize local information, known science, and future disease management.

## 2. Andean Lupin

### 2.1. Common Names

*Lupinus mutabilis*, in the family of legumes, is similar to *L. albus*, known as lupin in Spain; it is also known as bitter lupin. The local names in the Andean region are “chocho” in Colombia, Ecuador, and northern Peru; tarwi or tarhui in the Quechua language in cen-tral and south central Peru; tauri in the Aymara language south of Lake Titicaca in Peru and Bolivia, and chuchus muti in the Quechua area of Cochabamba [2]. The name ullus is used in different places in southern Peru and Bolivia. The name in English is “Andean lupin” or “pearl lupin” [7].

### 2.2. Origin

At least four thousand years ago, two ancient cultures in Egypt and the Andes were the first to domesticate and use species of Lupin in their diets: *L. luteus* in Egypt and *L. mutabilis* in the Andes. These species were used for similar food purposes. Curiously, both cultures first developed a maceration and washing processes to eliminate the alkaloids before using the seeds as food [8]. Remains of lupin seeds have been found in the tombs of the Nazca culture (100 to 500 BC) on the desert coast of Peru [9]. In the south of Peru, paintings depicting the lupin in ceremonial vessels from the Tiahuanaco culture (500–1000 AD) are an indication of its wide distribution [2].

### 2.3. Seed Yield Production

Lupin is characterized by low yields in the Andean zone. The national average yields are 0.40, 0.65, and 1.33 t/ha in Ecuador, Bolivia, and Peru, respectively [10,11]. Australia represents a special case, where currently more than 500,000 ha of sweet lupinus (*Lupinus angustifolius*) are cultivated mainly for use in animal feed [12]. The contribution of Australia to world production of lupin has averaged between 74 and 80% and the crop is produced in the grain belt of Western Australia [13]. Average yields in Australia and Western Europe exceed 1.5 t/ha [14].

It might be thought that low yields in the highland Andes are due to adverse conditions for plant growth, such as low temperatures, droughts, or soils deficient in nutrients, but this is not always the case. On experiment stations, as well as on high input farms where appropriate soil fertilization is used, Andean lupin yields often exceed 1.8 t/ha [11,15]. The large gap between achievable yields and current national averages suggests a tremendous potential for improving livelihoods through more profitable and sustainable production of Andean lupin cultivation.

### 2.4. Nutritive Value and Potential Pharmacological Use

After a debittering process, the Andean lupin grain is used for human consumption [16]. The nutritional composition of the cultivated lupins varies greatly among species. *L. mutabilis* contains the highest level of protein and oil in seeds [17]. Protein content varies from 41 to 51% on a dry weight basis [18,19]. The low proportion of testa (13%) in the seed of *L. mutabilis* could explain the high levels of protein in whole seeds compared with others lupin species [17]. Grain lupin protein compares favorably with other legumes, such as soybeans (Table 1). Lupin is sometimes referred to as “Andean soya” due to it has the potential for improving human nutritional standards in the Andean Highlands [17], and currently is also of high interest in European countries [20,21]. Oils and crude fiber contents are similar to that of soybean (Table 1). Oils in Andean lupin seeds are of high quality and appropriate for human consumption [22]. The fibre in whole seed, when finely ground, proved to be highly digestible in pigs and increased the digestible energy in pigs to higher than expected levels [22]. The specific carbohydrate composition of Andean lupin grain is related to anti-nutritional factors [12], characterized by low levels of starch and high levels of non-starch polysaccharides (NSP) (Table 1). In relation to the composition of amino acids, Andean lupin grain has lower content of the nutritionally essential, sulfur-containing amino acids (methionine and cysteine) [12] and, like most legumes, is deficient in tryptophan and tyrosine (Table 1). Therefore, it is important to consume lupin accompanied by a cereal such as corn, rice, or wheat [23] and even better in combination with quinoa, which results in an ideal food to combat malnutrition [2].

Ancestral knowledge was the starting point for a group of researchers who set out to isolate and characterize secondary metabolites in each of the fractions of this plant (leaves, flowers, stems, grains) [25] and validate its use in the pharmaceutical field [2]. Lupin alkaloids show a good antimicrobial effect against bacteria and fungi of clinical interest [26]. Lupin has been used to enrich wheat bread, and lupin-wheat bread has a texture quite similar to standard wheat bread, but the average content of protein, fat, and dietary fiber increased significantly, and its regular intake lowered hydrolytic and glycemic indexes of those who consumed it [27]. At a pharmacological level, frequent consumption of *L. mutabilis* grain resulted in improved metabolic control, decreased blood pressure, and increased HDL-cholesterol in type 2 diabetes-mellitus (T2DM) patients [28].

### 2.5. Andean Lupin in Livestock Farming Systems

In addition to its nutritional properties, the cultivation of lupin is very beneficial for the productive system of the highlands [3,19]. Its roots fix atmospheric nitrogen and therefore contribute to improved soil fertility. Because of this, lupin is part of the traditional systems of crop rotation of the Northern Altiplano, including: quinoa-lupin-potato, quinoa-lupin-barley, or quinoa-lupin-quinoa [6], without use of other fertilizers. Potato usually yields more and better quality. Lupin incorporates into the soil a nitrogen equivalent of 150 to 300 units, which in economic terms would be like providing 7 to 10 bags of urea per ha, and that is a very important contribution to small farmer incomes [29]. In addition, some soil-borne pathogens reduce their levels in soils due to a non-host lupin crop rotation [13].

## 3. Constraint

### 3.1. Foliar Diseases as the Main Constraint in Lupin around The World

Many constraints cause lupin yield instability in the Andean zone, including low soil fertility, water stress, and frost. However, lupin is susceptible to a wide range of diseases, among them foliar fungal diseases (Table 2), which are a major constraint to crop productivity. Anthracnose may appear less frequently than other foliar disease, however it is the most severe because it affects the crop at all developmental stages, causing lesions on stems that lead to typical twisting. Flower bud infection and lesions on young pods may result in complete loss of pods or infected seeds. Anthracnose develops in patches within the crop, spreading through rain splash of spores. Patches of deformed plants will form within the crop and these will coalesce and increase in size as the disease spreads after rains. If not managed, anthracnose can lead to total crop loss in susceptible varieties [30,31].

Anthracnose previously characterized as *Colletotrichum acutatum* [33,34,35], currently *Colletotrichum lupini* [36,37], is considered the most important disease in the Andean zone [34] and around the world [37]. Direct losses in the Andean countries have been estimated between 60 and 70% [38], which supports the general consensus of the importance of this disease.

### 3.2. Epidemiology and Disease Symptoms

Tropical highland areas of Ecuador, Peru, and Bolivia present weather conditions that are favorable for anthracnose to become more severe. Frequent rains, high relative humidity, and warm temperatures stimulate pathogen development and spread. Because of these favorable conditions and the fact that there are no resistant varieties, many fields are often completely destroyed [32,38].

Initial inoculum from focal sources contributes to epidemics [31,39]. Potential sources of inoculum are infected seed or volunteer plants from where it could spread from within or from other lupin fields [30,31,32,38]. On and below the seed coat, the pathogen develops compact dark intra- and intercellular mycelium [40], which transmit from the seed coat to seedlings during seed germination, colonizing the cotyledonal leaves, plumule, and radicle [41]. Rain splash spreads spores and the plant can be infected at any phenological stage [31,42]. Mycelium is the primary inoculum [30,31,32,37] and also the overwintering structure [31,37]. On infected plants, secondary inoculum is rapidly developed and dispersed through the crop by wind and rain splash [13,30,31,32]. Conidia are formed in all types of lupin tissues, including seeds, stems, and pods, in both wild and domesticated lupin species [32]. Black spots in the lesions corresponds to acervuli, which are asexual fruiting bodies formed during sporulation of the fungus [43]. Eventually, due to infections occurring along all developmental plant stages, seed yield is seriously affected [32,42].

The fungus attacks the stem (Figure 1A), producing necrotic spots and the typical main stem twisting. The attack continues on leaves (Figure 1B) and terminal shoots, destroying flower buds, seriously reducing grain production. Attacked pods show depressed lesions of 1 to 3 cm in diameter, covered with an orange layer, due to the fungi masses conidia (Figure 1C). The seeds have a “sucked” appearance in severe attacks (Figure 1D), whereas light infections do not cause these symptoms. Plants that grew from infected seed show symptoms on cotyledons and stems, sometimes killing the plant [43]. The initial amount of disease or a delay in the onset of disease can be lessened by reducing these sources of initial inoculum. Management programs based on the use of healthy seed and sanitation (elimination of volunteer plants) can improve disease control.

### 3.3. Disease Cycle

Anthracnose survives in or on infected seed and on old lupin trash. The pathogen does not survive in the soil in the absence of stubble residue, but it can survive up to two years in infected seeds [30]. Planting infected seed will produce infected seedlings that will rapidly produce lesions on roots, cotyledonal leaves, leaf petioles, or stems. Large numbers of spores will be produced in these lesions that, through rain, can splash on nearby healthy plants and establish the disease in new and surrounding plants. Anthracnose can survive for a short period of time on infected stubble, but the important source of inoculum is from infected seed. Rain stimulates the pathogen and spores produced in old stubble, which can infect new plants through rain splash and direct contact with self-seeded lupins (Figure 2) [30,31,32].

## 4. Methods to Eradicate the Pathogen from Infected Seed as the Primary Means of Pathogen Survival

### 4.1. Traditional Methods for Seed Disinfection

Infected seed is the primary inoculum for *C. lupini*. Unfortunately, in the Andean zone most lupin growers use their own seed, which is often infected [3,32]. Local farmers themselves collect the lupin seed. Due to the good price in the market, they prefer to sell commercial seed and keep small, wrinkled seeds with small reddish-brown spots for the next sowing [32]. Wrinkled and/or decolorized seed usually means that the seed is infected [31,32]. This practice makes it more likely that the next generation of plants will be anthracnose infected and is one of the main reasons for low productivity of lupin in marginal areas. The farmers need to be more aware of how important it is to use good quality seed and learn about the use of good methods for seed disinfection [32].

### 4.2. Alternative Methods for Seed Disinfection

Emergence of lupin is notoriously irregular in the Andean countries for several reasons, but mainly because the seed has poor sanitary quality and variable physiological development. The social context is an important factor in lupin anthracnose management in the Andean zone [32]. Farmers generally wait for a significant portion of plants to emerge before making a fungicide treatment, and by this time many plants have been exposed to inoculum. Researchers have treated seed with systemic fungicides prior to planting to solve the problem of early infection [3], but this generally does not protect plants after emergence [42]. The use of alternative methods, alone or in combination with chemicals, can offer a greater spectrum of action and higher efficiency in seed disinfection.

#### 4.2.1. Dry Heat

A valuable option to reduce fungal infections in seeds is dry heat treatment. Studies conducted in Australia have shown that *L. angustifolius*, *L. luteus*, and *L. albus* seed heat treated with 60–80 °C for a two-week period or less reduced lupin anthracnose infection to an undetectable level, and these treatments did not greatly reduce seed germination [44]. In *L. mutabilis*, treatments at 65 °C on infected seed for 24 h reduced disease incidence to undetectable levels. In addition, pathogen transmission from seed to seedlings was reduced by 75% with 8 h dry heat treatment (Table 3) [45].

Seed germination, however, may be reduced by dry heat treatment. Little effect on seed germination of *L. angustifolius*, was observed with 60 °C for one or two weeks, but the same temperature significantly affected germination of *L. luteus* and *L. albus* [44]. In *L. mutabilis,* exposure times of 12 h at 65 °C reduced germination [45], comparable with the results reported by [44], with reductions of seed germination on *L. luteus* after 60 or 70 °C treatment for one week. However, treatments of 65 °C for 8 h on *L. mutabilis* did not greatly affect emergence of seedlings [45], comparable with the results reported by [44] on *L. angustifolius* exposed for 4 days at 65 °C. Development of *L. angustifolius* cultivars resulted from an introgression of a large number of wild and external accessions within a breeding program in which selections were made for seed adapted to weather extremes [46]. In contrast, *L. mutabilis* cultivars were selected based on agronomic traits and adaptation to the cold environments of the tropic highlands. Seed coat thickness of *L. mutabilis* is 50% lower and saturated fatty acids are 60% less than in *L. angustifolius* [47,48]. These facts may confer less resistance of *L. mutabilis* to dry heat in comparison with *L. angustifolius* [45].

Interestingly, emergence of seedlings increased with dry heat treatments at 65 °C for 8 h in four *L. mutabilis* cultivars (Table 3) [45]. This seedling emergence promotion has also been observed for cotton seed dry heat treated for 8 h at 60 °C [49]. Treatment of Andean lupin seed with moderate dry heat exposure time at 65 °C reduces anthracnose infections and promotes seedlings emergence.

#### 4.2.2. UV-C Radiation

One of the natural components of sunlight is ultraviolet radiation (UV), which is divided according to its wavelength into long UV-A (315–390 nm), medium UV-B (280–315 nm), and short UV-C (100–280 nm). Within these subtypes, UV-C radiation is used as an effective biocidal agent for the disinfection of water, surfaces, and the air [50]. Similarly, seed infections can be reduced or eradicated by UV-C treatment. Treatment of anthracnose infected Andean lupin seed reduced infections by 90%, whereas doses of 86.4 kJ/m^2^ reduced infection to undetectable levels [51]. In other pathosystems, the mycelium of *Colletotrichum musae* was inhibited with 45 min of exposure to UV-C radiation, whereas treatments of 2 h reduced the diameter of the lesion in bananas inoculated with the same pathogen [52]. UV-C exposure times shorter than 20 min suppressed microbial decay on mangoes when the fruit received full irradiation [53]. Legumes, grains, and vegetables seeds are suitable for UV-C treatment. Optimal doses vary according to the species and are in the range of 0.025–480 kJ/m^2^ [54]. The direct germicidal effect of UV-C radiation could be the primary factor leading to disease suppression [55]. The control of anthracnose in lupin through the use of UV-C radiation is beneficial.

UV-C radiation may also reduce seed germination. Exposures for 24 h reduced germination of *Acacia ampliceps* seed from 80.1% to 65.2%. However, UV-C exposure of 6 h did not critically reduce germination [52]. In *L. mutabilis*, UV-C exposures for 8 h reduced seed borne anthracnose and had a slight effect on seed germination [51].

UV-C seed treatments enhance some physiological processes in seedlings. Andean lupin seedlings grown from UV-C pretreated seed had equal protein and chlorophyll con-tent compared to those grown from non-infected seed, and higher than seedlings grown from infected seed (Table 3) [51]. UV radiation doses stimulate the biosynthesis of UV-absorbing compounds and carotenoids, both of which perform a photoprotective role [60,61]. Red bean seedlings grown from seed pretreated with low doses of UV-C significantly increased protein content [62]. The plant response to environmental stress is given by the breakdown and recycling of proteins, which depends on the levels of proteolytic enzymes [63]. The pretreatment on lupin seed with UV-C for 8 h did not affect total chlorophyll and proteins, and doses of 57.6 and 86.4 kJ/m^2^ resulted in higher seed emergence and greater seedling dry weight. The positive effects were even better than that of fungicide seed treatment and contributed to further seedling establishment [51]. Oxidative stress in plants may be induced by UV radiation, increasing reactive oxygen species [63]. Several UV-absorbing compounds that activate antioxidant defense mechanisms are accumulated by plants to cope with this stress [64]. For instance, a significant increase in peroxidase and catalase activity was found in papaya peel as a result of UV-C treatment compared to initial values [65]. Components of cross-linking cell wall, polymerization of lignin and suberin monomers are associated with peroxidase enzymes and subsequent resistance in other host–pathogen interactions [66]. In the work of Falconi and Yanez [51], Andean lupin seedlings that grew from UV-C treated seed recorded higher enzymatic activity, similar to what occurs in tomato plants treated with azoxystrobin or *Pseudomonas fluorescens*, other treatments that induce plant resistance [67]. As a result of the research carried out by Falconi and Yánez [51] on the use of UV-C for treatment of *L. mutabilis* seed and its further effect on plant physiology and biochemistry, it is possible to hypothesize that UV-C on lupin seed acts as an inducer of systemic resistance. Low incidence of anthracnose observed in lupin seedlings grown under greenhouse conditions can be associated with changes in enzymatic activities [51].

#### 4.2.3. UV-B + Thermal Radiation

Solar radiation can efficiently reduce the survival of plant pathogens. Several types of fungi, such as *Pseudoperonospora cubensis* [68], *Peronospora tabacina*, *Uromyces phaseoli*, *Alternaria solani* [69], and *Phytophthora infestans* [70] have reduced viability as a result of exposure to solar radiation. The UV spectrum of sunlight causes most of the harmful effects on biological systems. Of the total solar radiation, approximately 8–9% corresponds to ultraviolet radiation. Although UV-B (280–315 nm) represents only 1.5% of the total UV spectrum, is of particular interest because it is the most active form [71]. Conidia of most species of phytopathogenic fungi directly exposed to UV-B radiation for a few hours can lose viability, but sub lethal doses of solar UV-B radiation can also delay conidia germination or reduce their virulence [72]. Differences in spore pigmentation may be associated with the effect of UV radiation on spore viability, as has been reported in other plant pathogenic fungi. For instance, spores of *Penonospora tabacina* were shown to be more sensitive to solar radiation then the more pigmented spores of *Alternaria solani* and *Uromyces phaseol*i, which supports the hypothesis that spore pigmentation affects sensitivity to solar radiation [73]. The effect of UV-B alone was determined by exposing Andean lupin seed lots at ambient temperature under an absorption UV-B film for 60 min. UV-B doses of 3.75 kJ/m^2^ reduced anthracnose infection seed by 60% [56]. Although *Colletotrichum* species can overwinter as mycelia under the seed coat [40], solar radiation reduced its viability [56].

Seeds of some plant species have seed coats that are more resistant to UV-B rays [74]. In this sense, depending on the plant species, low to moderate doses of UV-B may not re-duce seed germination, and may even increase it [75], whereas high doses of UV-B radiation can reduce seed germination [76]. In addition, solar radiation generates heat, and this radiated energy can also have an effect on disease [77,78]. However, seed exposure to UV-B radiation can induce a variety of responses in seedlings [79]. A solar oven was designed by Falconi and Yanez [56] to collect thermal radiation. Because it is unknown how protective the Andean lupin seed coat is against the sun’s extremely incandescent gases, the oven was covered with a film that partially absorbs UV-B. Thus, only the portion of UV-B that enters the solar oven will generate thermal radiation that will be transmitted to the infected seed (Figure 3).

Anthracnose infection in lupin seed was reduced more efficiently in the solar oven where the cumulative UV-B radiation and high temperature were combined, as compared to the individual effect of UV-B radiation at ambient temperature or of dry heat. On sunny days between 11 am and 2 pm, UV-B doses of 2.83 or 3.75 kJ/m^2^ were reached after 45 or 60 min, and temperature fluctuated between 73 and 79 °C in the solar oven; the ambient temperature varied from 19 to 23 °C. The high temperatures in the oven represent harsh conditions for seed borne anthracnose [56]. Anthracnose seed infection was reduced by 95% with UV-B doses of 2.83 kJ/m^2^ in the solar oven (Table 3), being more efficient than the isolated dry heat treatment at 65 °C for 60 min or the same UV-B dose of 3.75 kJ/m^2^ at ambient temperature. Viability of anthracnose and its subsequent potential plant infection were reduced with pretreatments of infected seed in the solar oven. Acervuli and mycelium of *C. lupini* living in the inner layer of the testa of lupin seed may be radically affected by active UV-B plus temperature [56].

The pretreatment of seed with UV-B irradiation can induce a variety of responses in seedlings and plants [80], but information about seed invigoration due to the combination of UV-B radiation with high temperature is scarce [80]. Combined UV-B doses and temperature achieved after 75 min in the solar oven decreased viability of *L. mutabilis* seed. However, moderate exposure times of 45 min, effective for seed borne disease control, do not reduce germination percentage [56]. Germination percentages of buckwheat, Chinese cabbage, turnip, cabbage, or parsley seeds were not greatly reduced with UV-B exposure times from 10 to 45 min [75]. New studies could elucidate the combined effect of UV-B and high temperature on seed invigoration and its potential expression into positive responses in the plant.

The synthesis of some heat shock physiological and biochemical processes in plants may be stimulated by pretreatment with UV-B thermal-heat of seed. No oxidative damage of chlorophyll and proteins was detected in Andean lupin seedlings grown from seed pre-treated for 45 min. Conversely, these exposure times enhanced peroxidase activity in seed-lings (Table 3) [56]. Diseases, high UV-B levels, and extreme temperatures can elevate the amounts of reactive oxygen species (ROS), including superoxide radicals and hydrogen peroxide [76]. These molecules are very destructive as they participate in many chemical reactions with lipids, proteins, and nucleic acids resulting, for example, in damage to cell membranes due to lipid peroxidation [77]. However, increased activity of antioxidant enzymes that serve as “sunscreens” may also be elicited as an acclimatization response to UV-B solar radiation [78]. For the elimination of H_2_O_2_, the antioxidant defense system uses peroxidase as an important component [79,80], purifying and participating in the polymerization of lignin and suberin monomers and other cross-linked cell wall components [81]. Dry heat pretreatment of cotton seed at 80 °C for 8 h enhanced physiological and biochemical process that have a photoprotective role for seedlings [49]. In another study using *Indigofera tinctoria*, peroxidase activity slightly increased with supplemental UV-B radiation for 2 h per day for 2 days [82]. In this sense, the combination of UV-B and the high temperature achieved in the solar oven designed by Falconi and Yanez [56] may have contributed to biochemical changes (Table 3). Future studies could elucidate whether such UV-B treatments of lupin seed enhance molecules implicated in host defense or plant growth development.

#### 4.2.4. Biological Treatment for the Seeds

Biological treatments to control anthracnose in seeds based on the use of microorganisms, such as *Bacillus* spp., have been described as effective. Studies conducted by Yanez et al. [57] demonstrated that several *Bacillus subtilis* strains collected from lupin fields strongly suppressed *C. lupini* on Andean lupin seed [57]. Furthermore, living cells, cell-free compounds, and lipopeptides obtained from these strains significantly reduced anthracnose on seeds (Table 3 and Figure 4A–D) [58,59]. The efficacy of these strains is due to their biological characteristic to produce antifungal lipopeptides that progressively inhibited the mycelial growth and conidial sporulation and killed the fungal structures of *C. lupini* (Figure 4D) [57,58,59].

Several studies found that the broad antimicrobial effect of *Bacillus* spp. against pre- and postharvest disease is due to production of powerful bioactive compounds. *Bacillus* spp. mainly produce cyclic lipopeptides from three families: fengycins, iturins, and surfactins [57,58,83,84,85,86,87,88,89,90]. For anthracnose, the biological control potential of native *B. subtilis* strains by the production of these lipopeptides has been demonstrated [57,58,59]. Living cells of *B. subtilis*, including vegetative cells and endospores, showed a strong biocide effect (Figure 4A) against *C. lupini*. Cell-free supernatant produced by these cells showed the same biocide effect as living cells (Figure 4B,C). Thin layer chromatography analysis (TLC) of *Bacillus* lipopeptide extracts demonstrated the presence of fengycin, iturin, and surfactin families [57,58,59] (Figure 4B). Furthermore, when TLC analysis was combined with the growth inhibition test in TLC-bioautography, the lipopeptide fractions corresponding to fengycin showed the highest antifungal activity against *C. lupini* when compared with iturins and surfactin fractions (Figure 4B). These results strongly support the role of lipopeptide production as the major antifungal factor used by *Bacillus* spp. [57,58,59] against *C. lupini*, similar to that described for other biocide compounds with antifungal activity against other *Colletotrichum* species in plants [87,88].

The biocontrol ability of native *B. subtilis* was confirmed by antifungal lipopeptide-related genes. PCR analysis, which showed fengycin E, iturin B, and surfactin B gene amplification associated with biocide fractions detected by TLC-bioautography, supported the role fengycins and iturins play as the major antifungal factors of *B. subtilis* native strains [58]. From a physiological and pathological perspective, the antifungal action of fengycins is due to their ability to interact with wall components of the fungal pathogen, such as ergosterol, and to alter its cell packing structure and permeability in a dose-dependent way [89] (Figure 4D). This mechanical damage on cell pathogen structures was reported by [88,89,90], who demonstrated that fengycins produced by *B. subtilis* were able to cause morphological alterations on a wide range of plant fungal pathogens by formation of a fengycin–ergosterol complex. Thus, the susceptibility of *C. lupini* to *B. subtilis* strains mediated by lipopeptides could be explained by the effect of fengycins on the target membranes, leading to the inhibition of mycelial growth, conidial germination, and subsequent pathogen death [58].

The efficacy of native *B. subtilis* strains to control anthracnose was evident when inoculated on *L. mutabilis* seeds, verifying that the *B. subtilis* strains have high antifungal activity, inhibit mycelial growth and conidia germination, reduce seed infection to undetectable levels, and increase lupin germination percentage (Table 3, Figure 5A) [58,59]. In addition, rhizobacteria trigger induced systemic resistance (ISR) of plants against pathogens through the production of several bioactive metabolites such as antibiotics, siderophores, and volatile compounds [90,91,92,93,94,95]. *Bacillus* spp. can elicit systemic resistance by inducing the synthesis of antioxidant defense enzymes [96,97,98]. Native *B. subtilis* strains also promoted the synthesis of catalase, peroxidase, and superoxide dismutase in Andean lupin seedlings, which are precursor substances of plant resistance (Figure 5B) [59]. *Bacillus* spp. increased synthesis of antioxidant defense enzymes and resulted in induced systemic resistance against other plant pathogens in tomato seedlings [94,95], and suppressed anthracnose disease of chili by the activation of defense-related enzymes [96]. Similarly, *B. subtilis* is capable of impairing disease incidence, promoting seedling growth, and increasing activities of antioxidant enzymes (POD, PPO, PAL) in cucumber plants [97]. Native *B. subtilis* promoted the growth of Andean lupin radicles and plumules in pre-inoculated seed (Table 3, Figure 5A) [58]. Plant hormone biosynthesis by *Bacillus* spp. has been directly related to subsequent growth promotion in different plants [94,95].

## 5. Strategies to Suppress Secondary Infection and Pathogen Spread

### 5.1. Biological Control in the Field

From a commercial point of view, the use of *Bacillu*s-based preparations has been widely demonstrated for efficiency controlling multiple field fungal pathogens [95]. However, research for the biocontrol of lupin anthracnose is scarce. Studies conducted in the field with anthracnose natural inoculum pressure showed that all *B. subtilis* treatments reduce the area under disease progress curve (AUDPC) compared with controls that showed large stem lesions, accompanied by necrotic tissue 90 days after sowing (DAS) and abundant sporulation, and in some cases dead plant from 105 to 120 DAS [99]. Lupin anthracnose can crack the stem at the site of infection (Figure 1A); multiple lesions can lead to stem twisting, or to complete stem collapse [30,31]. The AUDPC reduced with sequential applications of antagonists 1×10^9^ UFC/mL, every two weeks, compared to the AUDPC of untreated control in susceptible cultivars (Figure 6). In general, treatments with bacterial suspensions result in healthier stems, leaves, and pods, which are agronomic characteristics of lupin linked to production [99]. The significant reduction of AUDPC may be due to *B. subtilis* having a cascade effect on the different components of the disease triangle, that is, on the environment, the host plant, or the pathogen. Therefore, it can affect the onset and progress of the disease and also turn on resistance genes in the host [100]. *B. subtilis* induces disease resistance in the plant [93,95,96], or in turn promotes plant growth, a fact that makes it easier for plants to control pathogenic infections [90,92,97] or suppress diseases based on systemic resistance [93,94,95,97]. *B. subtilis* have also shown to control ginseng anthracnose, *Colletotrichum panacicola*, where leaf lesion diameter was not significantly affected compared with the fungicide treatment [101]. The protection mechanism of *B. subtilis* was associated with reduction of the incidence and with the initial processes of the infection cycle such as spore germination, appressorium formation, and penetration [100].

A *B. subtilis* strain selection program generally begins with screening under con-trolled laboratory or semi-controlled greenhouse conditions; however, it is difficult to predict how the bacteria will respond when released into the natural environment [103]. The native bacterial population on the phylloplane was estimated to confirm its protectant effect. Viable cell concentrations of *B. subtilis* were reduced in approximately 2.0 logarithm after 15 days of the first application on Andean lupin (Figure 7) [99] or reduced by 50% in strawberry leaves in the open field after 8 days of application [103]. Environmental stresses such as intense sunlight, dryness, or high temperature could reduce initial colonization [99,100], or rainfall could easily remove the bacterial population from the plant surface in several hours [103]. However, the population of *B. subtilis* will vary to some degree naturally once it has been established in the field. The average of native *B. subtilis* population on the phyllosphere of lupin remains stable at approximately 7.0 LOG after successive two-week-sprays for two months, which suggests that after the population has become stable, it can resist harsh environmental conditions (Figure 7) [99].

The total viable epiphytic population on the phylloplane can be determined by a serial dilution plating method considering that bacteria can colonize stomata, trichomes, vein endings, cell wall junctions, or even be beneath the leaf cuticle [104]. In the work of Falconi et al. [99], lupin stems and leaves sprayed with bacterial antagonists were consistently protected from natural infection of *C. lupini* fungi. Control was evident in a significant reduction in the AUDPC and suggests that native *B. subtilis* are more likely the cause of reduction in anthracnose severity. In addition, plants release organic compounds, such as sugars, organic acids, and growth regulators [105] that can stabilize cell concentration of *B. subtilis* due to nutrient availability [106]. Thus, when bacterial antagonist adhered, invaded, and survived on flowers, they effectively prevented the pathogen from colonizing the flowers [103]. Addition of 0.5% unrefined sugar to *B. subtilis* suspensions provides an initial food supply and increases adherence and colonization of bacteria in cocoa pods, improving monilia pod rot control [106,107]. Other studies show that a single inoculation of *B. pumilus* reduced banana sigatoka, *Mycosphaerella fijiensis*, by 33.6% and delayed progress of the disease by 21–28 days compared with the control [108]. Lower AUDPC values in comparison with the control indicates reduction of the initial inoculum or reduction of pathogen on the plant [108]. By associating prior findings [57,58,59] with the field observations where infections and disease severity were reduced, it can be speculated that biocontrol by native *B subtilis* resulted not only from damage of cellular structures (hyphae, mycelium, spores) of the pathogen, as occurred in the in vitro studies, but also by induction of systemic resistance (ISR) and promotion of plant growth mechanism (PGM). An innovative strategy for the integrated management of anthracnose in lupin could involve the activation of ISR by *B. subtilis*. Future studies could demonstrate more efficient use of these native strains and perhaps allow the establishment of biocontrol strategies.

### 5.2. Chemical Control

As there are no resistant varieties of Andean lupin and in the absence of any accurate method of controlling the disease, chemical control has been recommended as the most effective measure to reduce the spread of the disease. Synthetic fungicides reported for control of lupin anthracnose in the Andean zone and around the world are listed in Table 4. Seed treatment with fungicides reduces transmission of the pathogen from seed coat to seedling, but it does not provide complete control [109]. The synthetic fungicides generally recommended for controlling anthracnose disease are based on copper compounds, dithiocarbamates, benzimidazole and triazole compounds, manganese ethylenebisdithiocarbamate (Maneb) [109,110,111], and carbendazim [110], although the use of fungicides has been found ineffective under severe disease outbreak [109]. Newer chemicals like strobilurins (e.g., azoxystrobin, pyraclostrobin) have also been used for lupin anthracnose management [38,109,112]. Different classes of fungicides have specific modes of action and also differ in the duration of disease control. Farmers should take prevailing environmental conditions into consideration in their choice of fungicides. Rotation of two or more different classes of fungicides is highly recommended for increasing the chance of better protection against the disease in the fields [111,112,113,114]. Applications of Boscalid + Piraclostrobin or Azoxystrobin + Difeconazole significantly reduced the severity and infection in seed and significantly increased grain production [38].

Effective control through the use of chemical fungicides is possible by timely application during the critical period favorable for the onset of the disease [109,110,111,112,113]. Generally, fungicides should be applied for seed disinfection [109], at young seedling emergence, including early flowering or full bloom, and early pod formation to restrict the entry of the pathogen to the plant system [109,110,111,112,113].

Rotating fungicides with different FRAC codes can help delay development of lupin anthracnose resistance to fungicides. Code numbers on fungicide labels, called “FRAC” groups, can help to develop chemical rotations that delay fungicide-resistance (Table 4). The optimal application time is when the *C. lupini* population is small and contained, when weather conditions make the pathogen susceptible to fungicide, or when lupin is at the most susceptible growth stage (Figure 8).

*B. subtilis* for control of lupin anthracnose is also listed in Table 4. Application of sus-pensions based on *B. subtilis* [59,99,115] into a rotation with foliar fungicide containing mancozeb, chlorothalonil, or azoxystrobin [38,116,117] at sowing, plant emergence, early flowering, and early pod formation can better prevent transmission of the pathogen to the seedling and suppress anthracnose in the field. Using a biocontrol agent can reduce damage to human health and the environment.

### 5.3. Control Volunteer Lupins and Alternative Hosts

Lupin production in the tropical Andean highlands varies in the amount planted, which in general follows local rainfall patterns and prevailing prices. The highland tropics are also the home of several alternative hosts for anthracnose, including tamarillo (*Solanum betaceum*), that are geographically grown in neighboring lands with lupin and give the pathogen more opportunities to reproduce and survive [34]. Other studies also showed that *C. lupini* was able to cross-infect other host plant and *Colletotrichum* species from soybean also cross-infected lupin with varying degrees of virulence [118]. The plasticity of the interaction of *Colletotrichum* spp. with different hosts is significant. In addition, several species of wild lupin, such as lupin afelpado (*L. alopecuroides*), lupin rastrero (*L. sarmentosus*), miniature lupin or allpa chocho in Quechua language (*L. microphyllus*), or Quito’s chocho (*L. pubescens*) [119], grow in tropical highland areas and thereby act as a reservoir for pathogen growth during drier periods [35]. *L. cosentinii* can act as another source of infection in Western Australia, where control of naturalized populations is an important component for disease management [13]. In the Andean areas, high levels of inoculum could come from other susceptible cultivated or wild lupins and other alternative hosts in the surrounding areas; consequently, sanitation at the field or farm level is very important for anthracnose management [34].

### 5.4. Ontogenic Resistance

The ability of whole plants or plant parts to resist or tolerate disease as they age and mature is called ontogenic or age-related resistance. Levels of resistance that develop during aging of plant tissue may greatly reduce eventual disease severity and may even lead to escape from infection or immunity [120]. In lupin, anthracnose resistance is not equally expressed at all developmental stages. An ontogenic model was developed for three lupin species—*L. angustifolius*, *L. luteus*, and *L. albus*—and integrated into a decision support system for anthracnose management. The beginning of flowering and full bloom represent growth stages with high susceptibility [121]. Based on a series of in vivo assays involving Andean lupin genotypes, early cotyledonal stages (2–3 leaves) and the beginning of flowering (9–11 leaves) were susceptible to anthracnose, whereas the vegetative development stage (6–8 leaves) was resistant [102]. Cotyledonal and flowering stages *of L. mutabilis* were confirmed as susceptible to anthracnose through spectroradiometry and unmanned aerial vehicles [122]. This plant response of susceptible genotypes of Andean lupin in Ecuador [3] reflects the current situation of resistant and susceptible cultivars of *L. angustifolius* in Australia [109,123], where epidemics of lupin anthracnose progress more slowly in plots of intermediate-aged plants and dramatically increase during flowering. Some stage-specific resistance genes might not be expressed in early and late lupin developmental stages [102]. Targeting disease susceptibility during the life cycle of the Andean lupin plant will contribute to development of a more viable management option for production in areas of high anthracnose risk in Ecuador and maybe to more efficient lupin anthracnose management around the world. Early growth, beginning of flowering, and early pod formation [102,109,121,122] are the stages when lupin is most susceptible to anthracnose. An integrated ontogenic plan can be used during these major plant susceptibility periods for scheduling application of biological or chemical treatments.

### 5.5. Integrated Methods at Critical Phenological Stages

No single management technique has been found to efficiently control lupin anthracnose, therefore integration of techniques is essential. The first critical step is to reduce infection in seed (Figure 5) by using dry heat [45], UV-C radiation [51], or UV-B radiation [56]. Further protection can be added by combining physic treatments with other techniques, such as biological [57,58,59,94,98] or chemical control [38,109,110,111,112,113]. If one of the physic treatments of seed is used for high rainfall regions, it should be supplemented with foliar sprays to suppress pathogen dispersion in the field and subsequent infections. Application of fungicides on flowering and podding stages reduced incidence and increased yield of susceptible and resistant cultivars grown in high-risk areas [109].

Applications of *B. subtilis* theoretically prevent the establishment of the pathogen, interrupt its development, promote plant growth, and induce acquired resistance in lupin [59]. Because local results obtained under field conditions are consistent [99], bacterial suspensions at 1 × 10^9^ UFC/mL should be applied prior to pathogen infection, at sowing time (Figure 8). Success in controlling lupin anthracnose disease can be achieved with a sequential two-week application [99]. Shorter interval applications of *B. subtilis* are often recommended during the rapid growth stages of the plant to obtain a commercially acceptable product [115]. Special attention should be given to cotyledonal, early flowering, full bloom, and pod-filling stages (Figure 8) because lupin has been shown to be more anthracnose susceptible at these phenological stages [100,102,121,122].

Other technological components are needed to further reduce the use of synthetic pesticides. For this, it is necessary to develop *B. subtilis* at a semi-industrial or industrial level or to use the pathogen-inhibiting substances extracted from bacteria in rotation plans with chemical products. These rotation plans would include the interaction of *B. subtilis*-based biopesticides with protective and systemic fungicides to determine dose, frequency, and times of application [124]. A low-cost medium for *B. subtilis* lipopeptide production is available [125], and a spray drying formulation [126,127] has been developed to expand survival and reproduction of the antagonist under field environmental conditions. A plan that includes the rotation of biological fungicides with chemicals and their application during the most susceptible plant growth stages will greatly minimize infection in the field and reduce losses at harvest for a more sustainable production of Andean lupin, until resistant varieties are developed.

## 6. Conclusions

The treatment of Andean lupin anthracnose is a complex process due to its multifactorial origins. Protocols of this disease around the world include the use of resistant varieties for higher risk environments; however, resistant varieties are not available for *L. mutabilis*. Infection-free seed is the first key step to obtaining high production and good quality seed. Thus, seed should be tested for the presence and quantity of anthracnose infection. Because chemical seed dressing does not guarantee pathogen eradication, the use of alternative methods, such as dry heat, UV-C, and UV-B treatment is more effective for seed disinfection and reducing subsequent potential transmission to seedlings. These methods also induce seed invigoration that translate into positive physiological and biochemical responses in the plant. The use of *B. subtilis* for seed disinfection is a promising alternative. *B. subtilis* should be applied to seed at sowing time.

A second key aspect is to minimize secondary infections and spread of the pathogen in the field. A combination of alternatives is necessary to protect the crop and reduce yield losses due to anthracnose. Wild lupins and other alternant hosts that grow in surrounding areas of a lupin crop may be common sources of inoculum; if possible these plants should be destroyed. Certain phenological stages of Andean lupin have been identified as susceptible to anthracnose through bioassays and confirmed by using a multispectral lens of an unmanned aerial vehicle camera. It is crucial to reduce the initial processes of the infection cycle by using fungicides; rotating these with biological methods, such as *B. subtilis*, may be more effective in delaying the onset and slowing progress of the disease. Antimicrobial effect of *B. subtilis* lipopeptides may efficiently alter fungal chemical structure, enhancing disease control. Eventually, *B. subtilis* can stimulate plant and root growth and also turn on resistance genes in the host. *B. subtilis* should be applied at susceptible phenological stages alone or in a rotation plan with fungicides. Viable cell concentration of *B. subtilis* can be reduced after application on Andean lupin due to environmental stress; however, native bacterial populations became stable after several applications. New formulations will increase stability of cells and antimicrobial substances produced by *B. subtilis*. Future studies on application, colonization, and survival of native *B. subtilis* strains on the phylloplane will provide increasingly important components for establishing a sustainable program of integrated management of lupin anthracnose. Monitoring of anthracnose and other lupin stresses via geospatial technologies could provide a method to optimize resources for small farmers in underdeveloped regions.

## Figures and Tables

**Figure 1 plants-11-00654-f001:**
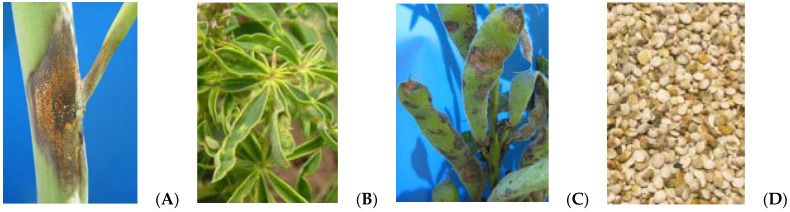
Characteristic symptoms of anthracnose on lupin stems (**A**), leaves (**B**), pods (**C**), and seeds (**D**).

**Figure 2 plants-11-00654-f002:**
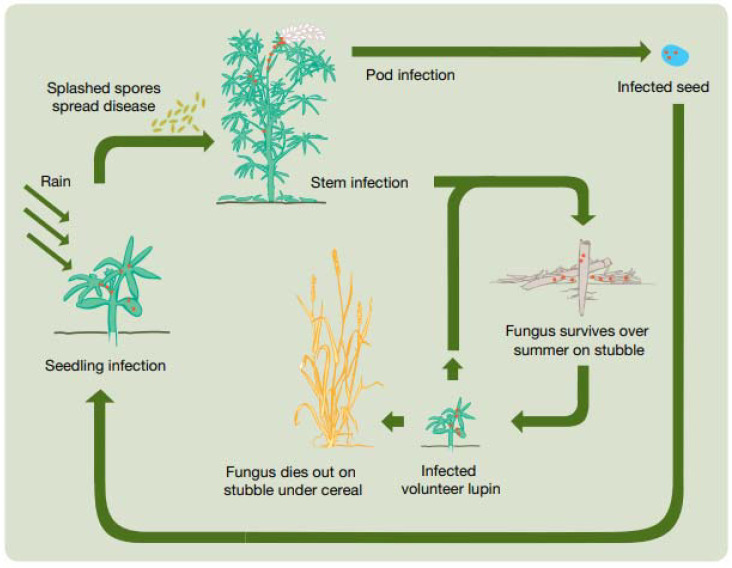
Life cycle of the anthracnose fungus (*Colletotrichum lupini*). Source: M Sweetingham [31].

**Figure 3 plants-11-00654-f003:**
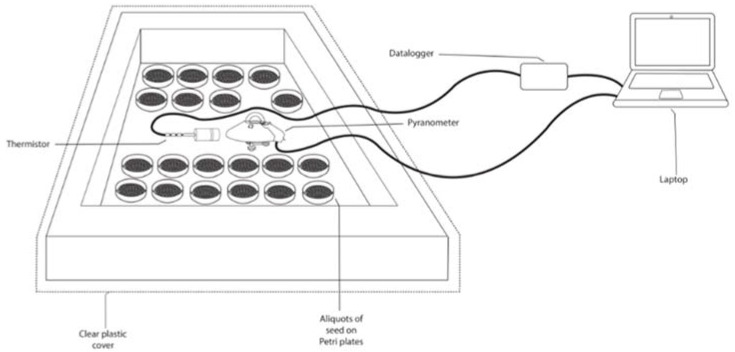
UV-B radiation solar oven schematic representation to treat Andean lupin seed lots infected with anthracnose. Source: [56].

**Figure 4 plants-11-00654-f004:**
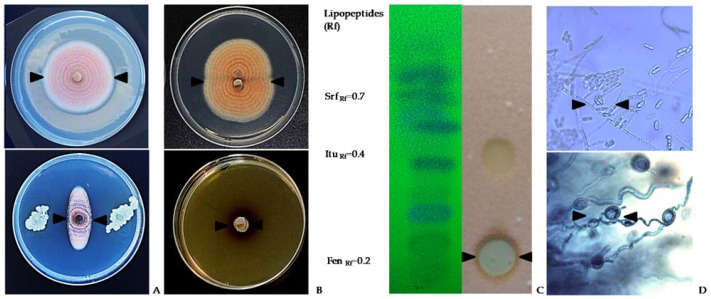
Antifungal activity of *Bacillus subtilis* against *C. lupini*. Black arrows show pathogen mycelial inhibition by bacterial biomass (**A**), cells free supernatant (**B**), and lipopeptides extract (**C**) compared with untreated control. Microscopic images of pathogen from growth zones (**D**) showing deformation and death of mycelia from bacterial treatments compared with untreated control. Source: [57,58,59].

**Figure 5 plants-11-00654-f005:**
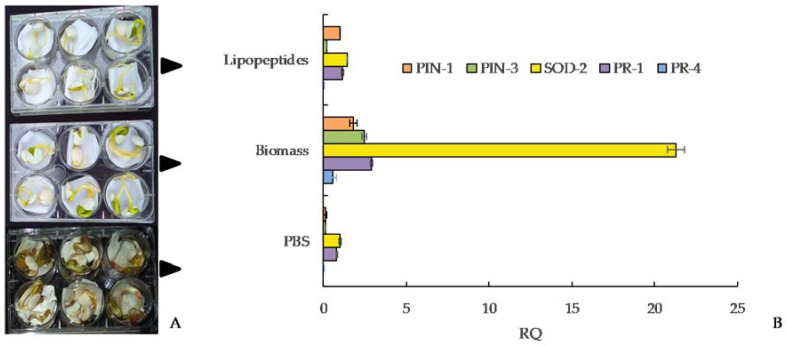
Effect of *Bacillus subtilis* treatments (**A**) in the expression of the *Lupinus mutabilis* growth-/ defense-related genes. (**B**) Relative quantity expression (RQ) of PIN-1, PIN-3, SOD-2, PR-1, and PR-4 genes induced by *B. subtilis* cells, lipopeptide extract, or sterile PBS as control was normalized against actin gene. Source: [59].

**Figure 6 plants-11-00654-f006:**
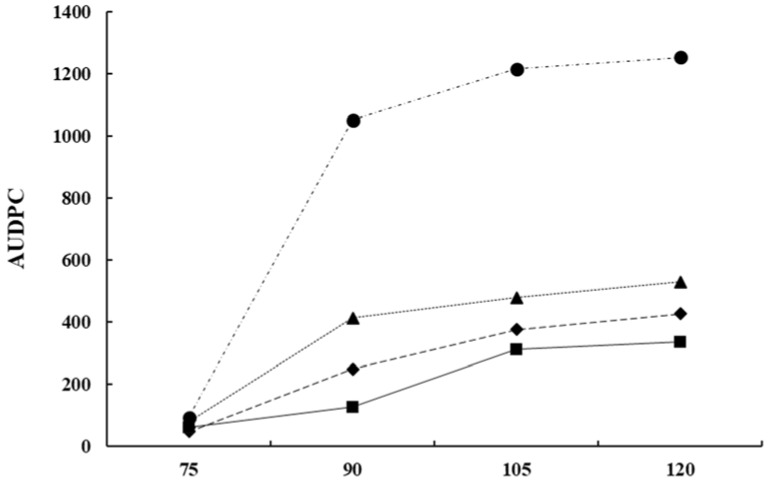
The area under disease progress 
curve (AUDPC) of anthracnose caused by *C. lupini* on the Andean lupin 
susceptible cultivar I.450 Andino. Natural pressure of inoculum represented as 
the control (Ck 

) and the AUDPC by effect of *Bacillus 
subtilis* Ctpx-S1 

, Ctpx-S2 

, and 
Ctpx-Z3 

. AUDPC calculates scores for 
lupin anthracnose severity on the 1-to-6 scale [102]. 
Pooled data from the 2015 and 2019 growing seasons at El Chaupi, Cantón Mejía, 
Ecuador, each point at disease onset plus 90, 105, and 120 days. Source: [99].

**Figure 7 plants-11-00654-f007:**
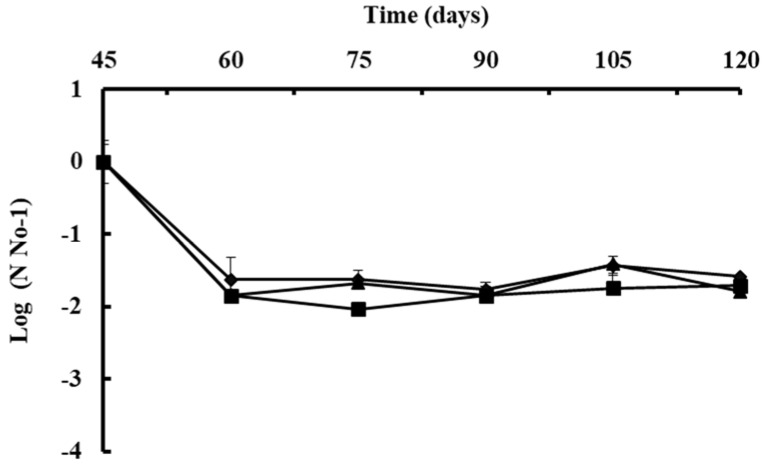
Population of *Bacillus 
subtilis* Ctpx-S1 

, Ctpx-S2 

, and 
Ctpx-Z3 

 recovered from the 
phyllosphere of lupin cultivar I-451 Guaranguito during 2015 and 2019 growing 
seasons. First spray at disease onset and fortnightly for eight weeks. Colony 
counted and results expressed as CFU *B subtilis* per gram of lupin 
leaves. To improve homogeneity of variances, data of bacterial concentration 
were log 10-transformed (log_10_ CFU/g). Each point represents the mean 
± SD of four independent repetitions of 10 g leaves. Source: [99].

**Figure 8 plants-11-00654-f008:**
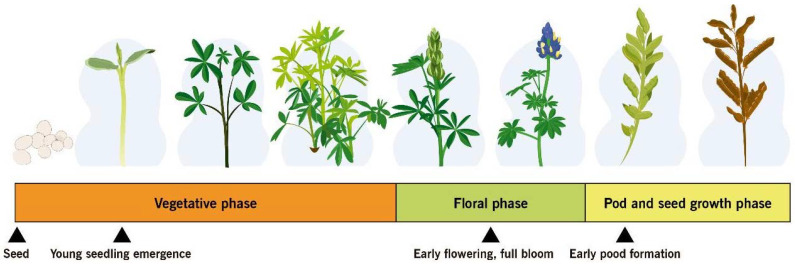
Anthracnose susceptible phenological stages of lupin.

**Table 1 plants-11-00654-t001:** Chemical composition of Andean lupin compared with soybean.

	*L. mutabilis* *	*Glycine max*
Crude protein (% of DM)	44.3	33.4
Oil (%)	16.5	16.4
Non-starch polysaccharides (%)	28.2	18.0
Crude fiber (% of DM)	7.5	6.45
Essential amino acids (g/16 gN)		
Glutamic acid	13.79	15.5
Aspartic acid	6.85	11.0

* Chemical content of lupin in the germplasm bank from the University of Cusco [2,24].

**Table 2 plants-11-00654-t002:** Main foliar fungal diseases of lupin (*L. mutabilis*) in the Andean zone.

Disease	Causal Agent	* Occurrence [2,3]	Symptoms and Importance
Anthracnose	*Colletotrichum lupini* [30,31]	8×	Stem bending, depressed lesions on pods, seed infection (Figure 1). Severe infections can result in complete crop failure [30,31,32]
Rust	*Uromyces lupinicola* [3]	21×	Brown spots and rust spores surrounded by lighter coloured halos on leaves. Partial defoliation in severe infections [3]
Ascochyta blight,	*Ascochita* sp. [30]	18×	Dark brown spots on leaves. Partial defoliation [30]
Phomopsis stem and pod blight	*Diaporthe toxica* [30,31]	18×	Usually on senescing lupin stems and pods. Disease symptoms while plants are green is uncommon. Minor crop losses [30,31]
Powdery mildew	*Erysiphe polygoni* [30,31]	n.r.	White floury covering comprising of conidia on leaves, stems and pods. Minor disease [30,31]

* Diseases variation from season to season, depending on pathogen presence, environmental conditions, and the lupin crop and varieties grown; n.r. not reported.

**Table 3 plants-11-00654-t003:** Alternative methods for the management of Andean lupin anthracnose infected seed.

Method	Doses—Exposure Time	Infection reDuction Efficiency	Additional Benefit on Lupin Plant
Dry heat	65 °C—8 h	75%	Promote seedling emergence [45]
UV-C	57.6 kJ/m^2^	90%	Enhance physiological and biochemical processes in seedlings [51]
UV-B + thermal radiation	2.83 kJ/m^2^ around 76 °C—45 min	95%	Seed invigoration, positive response in plant developed, increase activity of antioxidant enzymes [56]
*B. subtilis*	Living cells (LC) from 72-h-old cultures, 10^9^ CFU/mL or cell-free compounds (CFC)	76 to 100% (LC), 68 to 96 (CFC)	Positive effect on plant physiology, stimulate host defense and growth mechanisms [57,58,59]

**Table 4 plants-11-00654-t004:** Fungicides and bioproducts reported for the control of lupin anthracnose.

Active Ingredient	Chemical or Biological Group	Target Site *	FRAC Code **	References
Iprodione	dicarboximides	MAP/Histidine-Kinase in osmotic signal transduction	E3/2	[109,110]
Procymidone	dicarboximides	MAP/Histidine-Kinase in osmotic signal transduction	E3/2	[109,110]
Carbendazim	benzimidazoles	ß-tubulin assembly	B1/1	[109,110]
Difeconazole	triazoles	C14- demethylase in sterol biosynthesis	G1/3	[109]
Copper	inorganic	Multi-site contact activity	M 01	[109]
Mancozeb, Maneb	dithiocarbamates	Multi-site contact activity	M 03	[109,110,111]
Captan	phthalimides	Multi-site contact activity	M 04	[109]
Folpet	chloronitriles	Multi-site contact activity	M 05	[109]
Clorotalonil	chloronitriles	Multi-site contact activity	M 05	[109,111]
Dithianon	quinones	Multi-site contact activity	M 05	[112,113]
Ciprodinil + Fludioxonil	anilino-pyrimidines + phenylpyrroles	Multi-site contact activity	M 09	[113]
Fenhexamid + Tebuconazole	hydroxyanilides+ triazoles	3-keto reductase, C4- de-methylation + C14- demethylase in sterol biosynthesis	G3/17 + G1/3	[113]
Azoxistrobin	QoI-fungicides	Complex III: cytochrome bc1	C3/11	[109,110,111]
Pyraclostrobin + Boscalid	methoxy-carbamates + pyridine-carboxamides	Complex III: cytochrome bc1 at Qo site + complex II: succinate-dehydrogenase	C3/11 + C2/7	[38]
Azoxystrobin + Difeconazole	QoI-fungicides + triazoles	Complex III: cytochrome bc1 at Qo site + C14- demethylase in sterol biosynthesis	C3/11 + G1/3	[38]
*B. subtilis*	microbial	Multiple effects antibiosis, membrane disruption byfungicidal lipopeptides, lytic enzymes, induced plant defense	BM 02	[59,95,115]

* Biochemical mode of action given by the Fungicide Resistance Action Committee (FRAC). Fungicides groups are classified according to their risk level of developing resistance [114]. ** Numbers and letters used to distinguish the fungicide or biological groups according to their cross-resistance behaviour. FRAC codes organized by numbers and letters [114].

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
