# Peer review of "Available Strategies for the Management of Andean Lupin Anthracnose"

_plants, 2022, doi:10.3390/plants11050654_

Round 1

Reviewer 1 Report

This manuscript is a good attempt to gather all possible disease management strategies and review future prospects to eliminate/suppress the disease. Here are some comments regarding sentence formation, and there are many more changes to be made in terms of writing (almost in every paragraph), which is too much for me to mention. For example, the subheadings of the disease control strategies do not relate to what is described in the section. Additionally, this review does not show a disease management PLAN, but reviews different available disease management strategies. The title should be changed accordingly. There are numerous grammatical mistakes and sometimes the sentences do not mean anything. The manuscript needs thorough English editing.

Abstract

Line 12. …most devastating as it affects the whole crop including leaves, stem, pods, and seeds.

Line 17-19. Rephrase.

Line 20. This review covers some local …..

Line 37. Diseases have…quality of the seed, this review is a much-needed study to gather all the local information and science known to control…….

Line 53. Origin

Line 139. Constraint

Line 186. The sentence does not match the content. It mostly describes methods or various ways to eradicate the pathogen. Same for line 450, disease management for secondary infection.

Author Response

This manuscript is a good attempt to gather all possible disease management strategies and review future prospects to eliminate/suppress the disease. Here are some comments regarding sentence formation, and there are many more changes to be made in terms of writing (almost in every paragraph), which is too much for me to mention. For example,

The subheadings of the disease control strategies do not relate to what is described in the section.

ANSWER: Subheadings has been carefully checked in order to relate with the content of each section. Done.

Additionally, this review does not show a disease management PLAN, but reviews different available disease management strategies. The title should be changed accordingly.

ANSWER: Line 2, title changed. Done.

There are numerous grammatical mistakes and sometimes the sentences do not mean anything. The manuscript needs thorough English editing.

ANSWER: The manuscript was thoroughly revised by a native English speaker. Done.

Line 12. …most devastating as it affects the whole crop including leaves, stem, pods, and seeds.

ANSWER: Included in line 14. Done

Line 17-19. Rephrase.

ANSWER: Lines 17-19 were rephrased. Done.

Line 20. This review covers some local …..

ANSWER: Included in line 24. Done.

Line 37. Diseases have…quality of the seed, this review is a much-needed study to gather all the local information and science known to control…….

ANSWER: Included in lines 41-44. Done.

Line 53. Origin

ANSWER: Sub-heading revised. Done.

Line 139. Constraint

ANSWER: Sub-heading revised. Done.

Line 186. The sentence does not match the content. It mostly describes methods or various ways to eradicate the pathogen.

ANSWER: Agree. Methods to eradicate the pathogen from… line 203. Done.

Same for line 450, disease management for secondary infection.

ANSWER: Agree. Strategies to suppress the pathogen… line 448- Done.

Reviewer 2 Report

The MS must be improved overall at the beginning of its structure.

Particularly, the MS lacks of adequate description of disease framework of lupin caused by other pathogenic fungi and bacteria. This issue is fundamental for a comprehensive review as it can help readers to understand the scientific soundness of it. Why the authors have chosen the anthracnose rather than other fungal diseases in their review? Because other diseases are more or less widespread and/or because they are more or less severe than anthracnose? This section can be insert between the paragraphs 2 (Andean lupin) and 3 (Anthracnose) and supported by a table.

Second, the MS lacks of adequate description of the disease life cycle of Colletotrichum  spp. on lupin. This issue is fundamental for a comprehensive review as it can help readers to understand what are the critical stages (more pathogenic) of the pathogen in relation to the host infected (lupin) to understand the various strategies that can be implemented for its control both on the seed and on the adult plant in field as well described in the following two sections (4 and 5). This sub-paragraph can be insert after the sub-paragraph 3.2. (Epidemiology and disease symptoms) and supported by a figure.

Third, the MS lacks of adequate description of the traditional methods for seed disinfection. Only par. 4.1. is reported (Alternative methods….). Why? This new sub-paragraph (4.1.) can be insert before the sub-paragraph 4.2. (ex 4.1.).       

Author Response

The MS must be improved overall at the beginning of its structure.

ANSWER: The manuscript has been improved at the beginning and thoroughly revised throughout the text. Done.

Particularly, the MS lacks of adequate description of disease framework of lupin caused by other pathogenic fungi and bacteria. This issue is fundamental for a comprehensive review as it can help readers to understand the scientific soundness of it. Why the authors have chosen the anthracnose rather than other fungal diseases in their review? Because other diseases are more or less widespread and/or because they are more or less severe than anthracnose? This section can be insert between the paragraphs 2 (Andean lupin) and 3 (Anthracnose) and supported by a table.

ANSWER: Agree. Main foliar fungal diseases of lupin (L. mutabilis) in the Andean zone included in Table 2. Anthracnose as the most devastating diseases described in lines 130-135. Done.

Second, the MS lacks of adequate description of the disease life cycle of Colletotrichum  spp. on lupin. This issue is fundamental for a comprehensive review as it can help readers to understand what are the critical stages (more pathogenic) of the pathogen in relation to the host infected (lupin) to understand the various strategies that can be implemented for its control both on the seed and on the adult plant in field as well described in the following two sections (4 and 5). This sub-paragraph can be insert after the sub-paragraph 3.2. (Epidemiology and disease symptoms) and supported by a figure.

ANSWER: Disease cycle included in Figure 3 and explained in a sub-paragraph on lines 183-193. Done.

Third, the MS lacks of adequate description of the traditional methods for seed disinfection. Only par. 4.1. is reported (Alternative methods….). Why? This new sub-paragraph (4.1.) can be insert before the sub-paragraph 4.2. (ex 4.1.).       

ANSWER:  A new sub- paragraph 4.1.Traditional methods for seed disinfection inserted lines 204-212.

Round 2

Reviewer 1 Report

The manuscript seems a lot better after the changes made and the figures included. I still have a few minor comments to add.

Line 128. change to constraint

Line 381. biological treatment for the seeds.

Line 443. Strategies to suppress secondary infection and pathogen spread

Line 567. Alternative hosts

Author Response

The manuscript seems a lot better after the changes made and the figures included. I still have a few minor comments to add.

ANSWER: Both authors would like to thank for your valuable comments and corrections that have helped to improve our review. We incorporated the revisions and amendments in the manuscript.

Line 128. change to constraint

ANSWER: Changed. Done.

Line 381. biological treatment for the seeds.

ANSWER:  Done.

Line 443. Strategies to suppress secondary infection and pathogen spread

ANSWER: Done.

Line 567. Alternative hosts

ANSWER: Done.

Reviewer 2 Report

The MS results now improved basing on the reviewer's suggestions. Little adjustments should be still done during the editorial typesetting.

Author Response

ANSWER: Both authors would like to thank you for your valuable comments and corrections that have helped to improve our review.
